# Transcriptional Regulation of Amino Acid Transport in Glioblastoma Multiforme

**DOI:** 10.3390/cancers13246169

**Published:** 2021-12-07

**Authors:** Robyn A. Umans, Joelle Martin, Megan E. Harrigan, Dipan C. Patel, Lata Chaunsali, Aarash Roshandel, Kavya Iyer, Michael D. Powell, Ken Oestreich, Harald Sontheimer

**Affiliations:** 1Center for Glial Biology in Health, Disease and Cancer, The Fralin Biomedical Research Institute at VTC, Roanoke, VA 24016, USA; rumans129@gmail.com (R.A.U.); joelle.martin@alimerasciences.com (J.M.); meganh03@vt.edu (M.E.H.); 2Department of Neuroscience, University of Virginia School of Medicine, Charlottesville, VA 22903, USA; bpq2fc@virginia.edu (D.C.P.); hbe5rr@virginia.edu (L.C.); 3College of Agriculture and Life Sciences, Virginia Polytechnic Institute and State University, Blacksburg, VA 24061, USA; aarashmed2018@gmail.com; 4Roanoke College, Salem, VA 24153, USA; kviyer01@gmail.com; 5Department of Microbiology and Immunity, Emory University School of Medicine, Atlanta, GA 30322, USA; michael.duane.powell@emory.edu; 6Microbial Infection and Immunity, The Ohio State University College of Medicine, Columbus, OH 43210, USA; Ken.Oestreich@osumc.edu

**Keywords:** glioblastoma multiforme, SLC7A11, p53, glutamate

## Abstract

**Simple Summary:**

Glioblastoma multiforme (GBM) is a highly invasive brain tumor that typically has poor patient outcomes. This is due in part to aggressive tumor expansion within the brain parenchyma. This process is aided by assiduous glutamate release via the System xc- (SXC) cystine–glutamate antiporter. SXC is over-expressed in roughly half of GBM tumors where it is responsible for glutamate-mediated neuronal cell death and provides excess glutamate to fuel tumor-associated epilepsy. Available pharmacological inhibitors have some promise, although they lack specificity and have poor bioavailability. Therefore, identifying regulators of SXC may provide a superior avenue to target GBM. In this study, we identify tumor protein 53 (TP53) as a molecular regulator of SXC in GBM.

**Abstract:**

Glioblastoma multiforme (GBM) is a deadly brain tumor with a large unmet therapeutic need. Here, we tested the hypothesis that wild-type p53 is a negative transcriptional regulator of *SLC7A11*, the gene encoding the System xc- (SXC) catalytic subunit, xCT, in GBM. We demonstrate that xCT expression is inversely correlated with p53 expression in patient tissue. Using representative patient derived (PDX) tumor xenolines with wild-type, null, and mutant p53 we show that p53 expression negatively correlates with xCT expression. Using chromatin immunoprecipitation studies, we present a molecular interaction whereby p53 binds to the *SLC7A11* promoter, suppressing gene expression in PDX GBM cells. Accordingly, genetic knockdown of p53 increases *SLC7A11* transcript levels; conversely, over-expressing p53 in p53-null GBM cells downregulates xCT expression and glutamate release. Proof of principal studies in mice with flank gliomas demonstrate that daily treatment with the mutant p53 reactivator, PRIMA-1^Met^, results in reduced tumor growth associated with reduced xCT expression. These findings suggest that p53 is a molecular switch for GBM glutamate biology, with potential therapeutic utility.

## 1. Introduction

Glioblastoma multiforme (GBM) is a type of invasive, glial-derived brain tumor which take the lives of 3 out of 100,000 Americans every year [1]. In addition to its life-threatening diagnosis, GBM presents several comorbidities such as personality changes, visual field defects, headache, and tumor-associated epilepsy [2,3]. Even after surgical resection, radiation therapy, and chemotherapy, this aggressive standard of care only provides patients with a median survival rate of 15 months [4]. Although there are considerable efforts to sequence tumors to understand the common architectural landscape of GBM mutations [5], each patient’s response to therapy and disease progression differs. Therefore, by macroscopically finding targets in tumor behavior and invasion, treatments may assist a wider array of patients, increase the capacity for personalized medicine, and mitigate the spread of a disease usually caught in its later stages.

Our previous work identified high system x_c_^−^ (SXC) expression as a predictor for worse prognosis in GBM patients [6]. SXC is a cystine–glutamate antiporter that regulates excitatory neurotransmitter homeostasis in the brain and is causal of tumor-associated epilepsy [7,8]. The SXC inhibitor and FDA-approved drug Sulfasalazine (SAS) has shown promise in pre-clinical models [9], because it acutely inhibited seizures and chronically reduced tumor growth by 80% in vivo [10]. However, SAS has poor bioavailability and when metabolized, loses its ability to block xCT [11]. Therefore, identifying signaling pathways that regulate SXC are attractive for developing new GBM therapies. Recently, others have shown that p53 exhibits a non-canonical role by regulating SXC in systemic cancers [12].

*TP53* is one of the most frequently mutated genes in cancer, with a canonical role in cell cycle regulation. Recurrent mutations arise in the DNA-binding domain, rendering p53 dysfunctional and no longer a transcriptional guardian of the genome [13,14]. Although mutations more commonly occur in secondary gliomas compared to primary tumors [15], the downstream effectors of p53 in GBM are not well known. Given the need to discover therapeutic targets for GBM treatment and the relevance of SXC, we investigated the potential to target p53 as a regulator of SXC-mediated GBM biology. We found that p53 binds to the *SLC7A11* gene encoding SXC and suppresses its expression. Hence, p53 status is negatively correlated with SXC expression, and manipulating p53 in vitro and in vivo predictively alters SXC expression and function.

## 2. Materials and Methods

### 2.1. Animal Experiments

All animal experiments were carried out in accordance with Virginia Tech Institutional Animal Care and Use Committee (IACUC) guidelines. The Virginia Tech IACUC abides by the Guide for the Care and Use of Laboratory Animals and is accredited by the Association for Assessment and Accreditation of Laboratory Animal Care (AAALAC International). Mice were maintained in a pathogen-free barrier facility, housed in maximum groups of five, and subjected to a 12 h light/12 h dark cycle. For PDX tumor line propagation, 6–7-week-old, female athymic nude Foxn1nu mice were utilized (Envigo/Harlan, Indianapolis, IN, USA), as previously described [6]. For intracranial injection studies, 7–8-week-old male and female C.B-17/IcrHsd-*Prkdc^scid^* (SCID) mice (Envigo/Harlan) were utilized.

### 2.2. Cell Culture

Experiments were conducted using the GBM cell lines D54-MG (WHO IV, GBM, gifted by Dr. D.D. Bigner, Duke University, Durham, NC, USA) and U251MG and three patient-derived xenolines labeled GBM12, GBM14, and GBM22. Adherent cell cultures were maintained in DMEM/F12 (ThermoFisher, Waltham, MA, USA, #11320-082) with 7% fetal bovine serum (Aleken Biologicals, Nash, TX, USA). GBM14 and GBM22 were maintained in DMEM (ThermoFisher #21041-025) supplemented with B-27 Supplement without vitamin A, (ThermoFisher #12587010), EGF (ThermoFisher #PHG0311), FGF (ThermoFisher PHG0261), sodium pyruvate (ThermoFisher #11360-070), gentamicin (Fisher BW17 518Z), and amphotericin (Fisher #BP264550). GBM12 cells were maintained in a NeuroBasal (ThermoFisher #12348017) media supplemented with B-27, EGF, FGF, L-glutamine (ThermoFisher #21041-025), amphotericin, and gentamicin. Cultures were incubated at 37 °C and in 10% a CO_2_ atmosphere. Brightfield images of cell cultures were captured on a Nikon Eclipse TS100 (Melville, NY, USA) inverted microscope.

### 2.3. Immunocytochemistry (ICC)

To maintain xenoline sphere cultures for ICC, cells were adhered to treated glass coverslips: 12 mm glass coverslips in a 6-well plate were incubated with 1:10 poly-L-ornithine solution (Sigma, St. Louis, MO, USA, #P4957-50 mL) diluted in PBS for at least 1 h at 37 °C. Coverslips were washed a few times with PBS before plating 200,000–400,000 cells per well. Cells were allowed to attach overnight before drugging. After appropriate time points were reached, cells were rinsed once in PBS and fixed in 4% paraformaldehyde at room temperature for 10–15 min. Coverslips were rinsed again with PBS, and then once briefly with PBSTw (PBS + 0.01% Tween 20). Following fixation and rinsing, coverslips were blocked with 10% donkey serum made in PBSTw for at least one hour at room temperature. Coverslips were carefully transferred to a humidity chamber for antibody staining. The following primary antibodies were diluted in blocking buffer and used for staining for 1 h at room temperature; p53: mouse anti p53 (DO-1 clone, Santa Cruz, Dallas, TX, USA, #SC-126,1:100) and xCT: goat anti xCT (Abcam, Boston, MA, USA, #ab60171,1:125). Following primary antibody incubation, coverslips were rinsed 6 times in PBSTw for 30 min. After washes, coverslips were incubated for 1 h at room temperature in the following secondary antibody combination diluted into blocking buffer: DAPI (1:1000, Fisher/LifeTech, Waltham, MA, USA, #D1306), Alexa Fluor« 488-AffiniPure Donkey Anti-Mouse IgG (Jackson ImmunoResearch, West Grove, PA, USA, # 715-545-150, 1:1000), and Alexa Fluor^®^ 488 AffiniPure Donkey Anti-Goat IgG (Jackson ImmunoResearch, West Grove, PA, USA, # 705-545-147, 1:1000). Coverslips were washed 6 times in PBSTw for 30 min and then mounted (Electron Microscopy Sciences, Hatfield, PA, USA, #17984-25) onto charged glass slides (VWR #48311-703).

### 2.4. Confocal Imaging

A Nikon AI confocal microscope was used for imaging ICC experiments. A 20x objective was used with a 3-times optical zoom setting for confocal z-stacks in the Nikon Elements software. Maximum intensity projections were created for all images.

### 2.5. DNA Isolation

Between 1 and 2 million cultured xenoline cells were collected in PBS and DNA was isolated using the QIagen DNeasy Blood and Tissue Kit, Germantown, MD, USA, (#69504), following manufacturer’s instructions for cultured cells. Samples were re-suspended in Buffer AE for further use in polymerase chain reactions (PCRs) and Sanger sequencing assays. DNA samples were quantified with a Nanodrop to ensure successful DNA extraction.

### 2.6. TP53 PCR and Sanger Sequencing

Thirteen hotspot regions, determined by the International Agency for Research on Cancer (IARC), were amplified using a slightly modified version of the IARC human *TP53* PCR 2010 protocol. The PCR regions amplified, primer sequences, programs, clean up, and alignment procedures for each reaction were used as previously published [16]. The sequenced genomic regions spanned exon 2 to 11, including splice junction sites, in order to minimize the number of missed mutations. All sequencing reactions were performed through Virginia Tech’s Genomics Sequencing Center.

### 2.7. Protein Quantification and Western Blot

For endogenous p53 and xCT expression, 1–2 million cells were harvested from cell culture by centrifugation. Samples were re-suspended in 200 µL cold radioimmunoprecipitation assay buffer (RIPA) buffer supplemented with protease (Sigma #P8340) and phosphatase (Sigma #P0044) inhibitors. Samples were sonicated twice in a cold cup horn with ice water and then centrifuged at 12,000 rpm for 20 min at 4 °C. Supernatants were collected for total protein concentration analysis with a Pierce BCA Protein Assay Kit (Thermo Fisher #23225). For Western blot analysis, 10–15 µg of cell or tissue lysate was prepared with 4X Laemmli sample buffer (BioRad, Hercules, CA, USA, #1610747), 50 mM Dithiothreitol (DTT) (Sigma #D0632-5G), and RIPA for each lane. Samples were further denatured on a heat block at 90 °C for 8 min before loading. Depending on sample numbers, 10–15 µg of each sample was loaded per lane of a 4–15% Mini-PROTEAN TGX Precast Protein Gel (BioRad #4561086) or MIDI 4–20% Criterion Precast Midi SDS page gel (BioRad #5671095), and electrophoresis was performed at 120 mV for 70 min. Protein was transferred to PVDF membranes using the Trans-Blot Turbo Transfer system (BioRad #1704150) and Turbo kit (BioRad #1704275). Membranes were blocked in blocking buffer (LICOR # 927-80001) for 1 h at room temperature. Blocked membranes were incubated with a mouse anti-p53 antibody (Santa Cruz Biotechnology, Dallas, TX, USA, #SC-126, 1:200) or goat anti-xCT antibody (Abcam #ab60171, 1:450) and chicken anti-GAPDH antibody (Sigma #AB2302, 1:5000) overnight at 4 °C. The membrane was washed the next day with TBST (0.1% Tween 20) for at least 30 min and incubated in blocking buffer with 0.1% Tween 20, 0.01% SDS, IRDye^®^ 800CW Donkey anti-Mouse (Li-COR, Lincoln, NE, USA, #925-32212, 1:20,000) or IRDye^®^ 800CW Donkey anti-Goat IgG (Li-COR # 926-32214, 1:20,000), and IRDye^®^ 680RD Donkey anti-Chicken (Li-COR #926-68075, 1:20,000) secondary antibodies for 1 h at room temperature in the dark. Membranes were washed again in TBST for at least 30 min at room temperature, briefly rinsed in TBS, and were imaged on a Li-COR Odyssey Fc Imager. Densitometry of individual bands was performed in FIJI/Image-J software (https://imagej.net/software/fiji/downloads) (accessed on 3 December 2021).

### 2.8. RT-qPCR

Total RNA was extracted according to the manufacturer’s instructions with Trizol (Thermo #15596026) and the PureLink RNA Mini Kit (Thermo #12183025). DNase (Thermo #12185010) was used to extract DNA-free RNA. Following extraction, cDNA was synthesized with SuperScript VILO Master mix (Thermo #11755500). Around 100 ng cDNA was used for each PCR reaction. Pre-designed TaqMan^®^ Gene Expression Assays were ordered for the following human genes: SLC7A11 (Hs00921938_m1), CDKN1A/p21 (Hs00355782_m1), TP53 (Hs01034249_m1), and IPO8 (Hs00914057_m1). All target genes were amplified with the FAM-MGB reporter, and IPO8 was used as an internal control with the VIC-MGB-PL reporter. A RT-qPCR master mix was prepared for each target of interest, according to manufacturer’s instructions, with TaqMan^™^ Universal Master Mix II, no UNG (ThermoFisher #4440040); then, the cDNA template was added. Reactions were plated in triplicates on 96-well reaction plates (ThermoFisher/LifeTech #4346906) and transcripts were amplified for 40 cycles using an Applied Biosystems StepOnePlus^™^ Real-Time PCR System (Waltham, MA, USA).

### 2.9. Database Mining

Correlation analyses between p53 and SLC7A11 expression was performed using the GlioVis database [17]. After entering the genes of interest into the platform (Gene 1 as TP53 and Gene 2 as SLC7A11), along with the outside database, sequencing platform, and histology (TCGA_GBM, HG-U133A, and GBM), the plots were then analyzed for statistical significance using Pearson’s correlation methods using GraphPad Prism Software 8.1.1. (SanDiego, CA, USA). The ‘r’ value (‘estimate’) and the number of samples (‘parameter’) were shown through the results from the platform.

For IVY Glioblastoma Atlas Project (IVY GAP) [18] Database mining, under the RNA-Seq tab, a “Gene Search” was performed for “GBM” and the following individual “Select Features”: Leading Edge, Infiltrating Tumor, Cellular Tumor, Perinecrotic Zone, and Microvascular Proliferation. All tumors were selected and filtered by the gene name “TP53” (Gene ID 7114) and “SLC7A11” (Gene ID 23410). Both panels were highlighted, and the data were downloaded for each select feature. According to the instructions in the “Contents” text document, the RNA-Seq expression values corresponded in pairs to the donor ID in the columns file. Values were calculated with z-score normalization through the database.

### 2.10. Chromatin Immunoprecipitation (ChIP) and PCR

The ChIP assay performed in this study was adapted from previously reported methods and with assistance from members of the Oestreich laboratory [12,19]. In brief, chromatin was harvested from GBM12, GBM14, and GBM22 cells and immunoprecipitated with 5 µg of antibodies to p53 (DO-1) (Santa Cruz sc-126X) or a mouse IgG control (Santa Cruz SC-2025). Roughly 10 ng/µL of precipitated DNA was analyzed with gene-specific primers (SLC7A11 forward: 5′- AGGCTTCTCATGTGGCTGAT -3′, and reverse, 5′- TGCATCGTGCTCTCAATTCT -3′; p21 forward: 5′- CTTTCACCATTCCCCTACCC -3′, and reverse. 5′- AATAGCCACCAGCCTCTTCT -3′; human control HSPA6 forward: 5′- AGGAGAGGACTTCGACAACCG -3′ and reverse, 5′- CAGGTCCTTCCCATGCTTCC -3′) by RT-PCR with GoTaq (Promega M7123), according to the manufacturer’s instructions and as previously detailed [12]. For each RT-PCR reaction, 10 ng of DNA was amplified and analyzed on a 1% agarose gel. Images were captured with the Azure c500 system (Dublin, CA, USA).

### 2.11. siRNA Transfection

GBM14 cells (wild-type p53) were nucleofected with 2 nM p53 (Dharmacon, (Cambrige, England, #L-003329-00-0005) or 2 nM control siRNA (Dharmacon #D-001810-10-05) using the A-033 program on a Lonza 2B Nucleofector. Cells recovered for 15 min in 950 µL of warm DMEM complete before plating. If multiple transfections were performed, transfections of the same target were pooled together before plating. Cells were harvested 72 or 96 h post-transfection.

### 2.12. Lentiviral Transduction

To over-express p53 protein in p53-null GBM cells (GBM12), cells were incubated in Accutase (Sigma #A6964-100ML), counted on a hemocytometer, and seeded at 250,000 cells per well of a 24-well plate. GBM12 cells were transduced with either a control lentivirus that expressed green fluorescent protein (GFP) (Origene; (Rockville, MD, USA, #PS100093V) or an experimental lentivirus that over-expressed p53 conjugated to GFP (Origene #RC200003L4V). Lentiviruses were used at a multiplicity of infection (MOI) of 2. Transduced cells were selected with two, 48 h treatments of 5 µg/mL puromycin (Sigma #P8833-10 mg). After puromycin selection, DMEM was replaced with NeuroBasal media, which does not contain glutamate. Cells incubated in NeuroBasal media for 24 h before collecting the media for glutamate measurements. Cell pellets were collected in RIPA buffer for subsequent protein quantification and Western blot analyses.

### 2.13. Glutamate Release

For siRNA experiments, cells were transfected and incubated for 72 or 96 h until a media change to NeuroBasal complete (no glutamate present). For lentivirus experiments, cells were transduced and maintained in culture for 5 days until a media change into NeuroBasal complete. Subsequently, 24 h post-media change, cells were collected in RIPA buffer and the medium from each cell pellet was collected for glutamate measurements. Total protein was measured as described in the methods above. Glutamate was measured using a Glutamate Assay Kit (Sigma-Aldrich-MAK004), according to the manufacturer’s instructions.

### 2.14. PRIMA-1^Met^ Flank Tumor Experiments

6–7-week-old, female, athymic nude mice were injected with GBM22 tumor cells (mutant p53 R273C), as mentioned above in the “Animal Experiments” methods. Throughout the duration of the experiment, mice were weighed and tumors were measured on at least 3 non-consecutive days. Tumors were allowed to establish for 7 days until animals were randomly assigned to one of two retro-orbital (RO) treatment groups: (1) sterile saline, or (2) 100 mg/kg PRIMA-1^Met^ (Cayman Chemical, Ann Arbor, MI, USA, #9000487), as used in previous experiments [20]. Animals were given daily RO injections for 5 days on, then 2 days off, then another 5 days on before humane euthanization.

### 2.15. PRIMA-1^Met^ Intracranial Tumor Experiments

GBM22 tumor cells (2.0 × 10^5^) were intracranially implanted in 7–8-week-old male and female SCID mice. Anesthesia was maintained at 1–3.5% isoflurane. Pre-operatively, buprenorphine (0.05–0.1 mg/kg) or buprenorphine SR (0.5–1 mg/kg), in conjunction with a dose of carprofen (5 mg/kg), was administered subcutaneously to prevent post-surgical swelling, inflammation, and pain. Depilatory cream was applied to the scalp hair to facilitate removal and the skin was swabbed with povidone–iodine solution and 70% alcohol in three alternating sets of application. A single one-centimeter incision was made, front to back, using a No. 11 scalpel blade (Feather Safety Razor, Osaka, Japan). The animals were set up on a stereotaxic apparatus, and 0.5 mm burr holes were bored through the skull on the left side 1.0–2.0 mm and 0.5–1.0 mm posterior from the bregma using a dental drill equipped with a 1.0 mm drill bit held by one arm of the stereotactic apparatus. A needle attached to a Hamilton syringe containing the glioma cell solution was inserted into this hole and the depth of needle insertion was measured using gradations on the stereotaxic apparatus holding the needle and micro-injector. The needle was lowered 0.5 mm deeper than the final injection depth and then slowly retracted to create a pocket. Then, 2–5 µL of the cell solution was injected slowly and continuously using an automated injection pump (UMP3-1 UltraMicroPump micro injector, World Precision Instruments, Sarasota, FL, USA). The needle remained in place for 2 min before it was slowly removed. Bone wax was used to seal the hole in the skull and the incision was closed using skin glue. The animals were given lidocaine topical ointment at the incision site for post-operative pain and returned to the cage placed on a warming pad. Animals were kept on warming pads and monitored until full consciousness was regained before returning to the main colony room. The incision site and overall health status of the animal was checked daily. After 7 days of tumor growth, animals were randomly assigned to either the sterile saline or 100 mg/kg PRIMA-1^Met^ treatment group for RO delivery. Animals were monitored daily, and a projected humane endpoint was reached around 16 days post-tumor implantation. Animals were humanely euthanized, the brain was dissected, and the tumor tissue was punched and flash frozen for further Western blot analysis.

### 2.16. Statistics

All graphs and statistical analyses were performed using GraphPad Prism software to determine *p*-values, where *p* < 0.05 was considered statistically significant. Data were analyzed with the Student’s t-test or ANOVA with a Tukey’s (one-way ANOVA) or Šidák’s (two-way ANOVA) multiple comparisons post hoc test.

## 3. Results

### 3.1. p53 Functional Status Is Relative to SLC7A11 Expression in GBM

The central hypothesis of this study was that p53 may act as a transcriptional regulator of *SLC7A11,* whereby the loss or mutation of p53 in brain tumors should show reduced *SLC7A11* expression as well as reduced cystine–glutamate transport activity. We previously characterized a comprehensive set of patient-derived glioma samples that were maintained as PDX lines. These studies segregated gliomas into low and high SXC expressers [6] with distinct biological differences. In the present study, we subjected these same PDX lines to Sanger sequencing and identified representative examples that have wild-type p53 (GBM14), mutant p53 (GBM22), and p53-null (GBM12) (Figure 1A). When probed for *SLC7A11* transcript and xCT protein, these lines showed the hypothesized inverse relationship between p53 expression and *SLC7A11*/xCT (Figure 1B,D). Note that through transcript and protein analyses, the mutant p53 expressing GBM22 exhibited increased p53 expression (Figure 1C). This is due, in part, to the well-known loss of p53 autoregulation and mutant p53 accumulation in human cancers [21,22]. To confirm that mutant p53 lost its canonical functional, we probed for p21, which is a positively regulated target of p53 [23]. Through RT-qPCR and Western blot, we found that only wild-type GBM14 cells had high levels of p21 transcript and protein (Figure 1D).

Through follow-up ICC experiments, we confirmed robust whole-cell xCT expression in GBM12 and GBM22 cells compared to GBM14 (Figure 1E). Although RNA and global protein analyses showed the highest p53 levels in the GBM22 line (Figure 1C), ICC revealed this higher p53 expression was retained in the nucleus, indicative of over-expression and of a mutant p53 status [24] (Figure 1F). Based on our sequencing, transcript, and protein analyses, we concluded that glioma cells bearing wild-type p53 result in lower levels of *SLC7A11*/xCT expression.

To expand the applicability of an inverse relationship between p53 and *SLC7A11*, we mined GBM patient sequencing data available through the GlioVis software portal and the TCGA study data [5]. We found a significant, albeit small, inverse relationship between *TP53* and *SLC7A11* expression (Figure 2A). Notably, these data may be skewed by misrepresentation of the true mutational status of p53 when relying on IHC analysis rather than Sanger sequencing [16]. We pursued an additional database, IVY GAP, to look at *TP53* and *SLC7A11* expression in different GBM tumor regions. Except for the microvascular proliferation zone, *TP53* and *SLC7A11* expression was significantly different in the cellular tumor, leading edge, and perinecrotic zone regions of GBM tumors, and once again showed an inverse correlation (Figure 2B).

### 3.2. Wild-Type p53 Is a Transcriptional Repressor of SLC7A11 Expression in Glioma

We next sought to determine whether there was a physical interaction between p53 and the transcriptional start site in the *SLC7A11* promoter. To do so, we performed ChIP assays that were designed according to previous experiments that demonstrated an interaction between p53 and *SLC7A11* in U2OS cells [12]. Our ChIP experiments showed p53 binding to the *SLC7A11* promoter region in wild-type p53 cells (GBM14), but failure to bind to the *SLC7A11* promoter in GBM12 (p53-null) and GBM22 (R273C) cells (Figure 3A). Notably, in GBM22, the observed binding was equal to the IgG control. A PCR for p21, which is a downstream p53 target and serves as a positive control for p53 interactions, confirmed that only GBM14 cells expressing wild-type p53 exhibit p21 binding (Figure 3A). These results support the conclusion that wild-type p53 physically binds to and regulates *SLC7A11* expression in GBM.

### 3.3. p53 Is a Molecular Toggle for SXC Glioma Biology

The major role of xCT in glioma is to import cystine for glutathione (GSH) synthesis and release glutamate to inflict excitotoxicity [25]. Previous studies were able to pharmacologically manipulate xCT functions [6,10]. We sought to determine whether manipulating p53 levels would similarly affect *SLC7A11*/xCT expression and function. We hypothesized that *TP53* knockdown via siRNA would increase *SLC7A11* expression and xCT function. RT-qPCR revealed the successful knockdown of *TP53* transcript (Figure 3B), resulting in increased *SLC7A11* transcript levels 72 h post-transfection (Figure 3C). However, this did not result in decreased xCT protein levels, even though p53 protein was reduced by 66.4% and 100% relative to control siRNA after 72 and 96 h post-transfection, respectively (Appendix A). Western blot analyses of the whole cell lysate would not discern xCT membrane expression versus transporter being actively trafficked in the cytosol; therefore, we next measured glutamate release, which would indicate xCT membrane activity directly from cells with reduced p53. These experiments did not show a decrease in glutamate release from cells with decreased p53 levels 72–96 or 96–120 h post-transfection (Appendix A). Hence, whereas *SLC7A11* transcript was altered after *TP53* siRNA knockdown, xCT protein turnover appeared to be slow, as previously shown [26], preventing the rapid downregulation of xCT function using p53 siRNA (Appendix A).

To show that xCT protein is regulated by p53 status, we took the opposite approach to over-express wild-type p53 in p53-null GBM12 PDX cells. For these experiments, we hypothesized that by over-expressing p53 in p53-null tumor cells, we would be able to decrease xCT expression and glutamate release. To do so, we transduced GBM12 cells with lentiviruses that either over-expressed GFP or p53 conjugated to GFP. After successful transduction (Figure 4A), cells were assessed for p53 and xCT protein levels. Western blot revealed the over-expression of p53 in GBM12 cells (Figure 4B) which was associated with a reduction in xCT protein in these cells (Figure 4C). Furthermore, media collected from the same samples had reduced glutamate levels compared to control media (Figure 4D), suggesting that restoring p53 in p53-null GBM cells helped mitigate SXC properties that make these GBM12 tumors so invasive in vivo [6].

### 3.4. PRIMA-1^Met^ Mitigates GBM Tumor Burden and xCT Expression In Vivo

We could manipulate p53 to mitigate in vitro tumor cell glutamate release; therefore, we wanted to examine whether this may be used as a therapeutic strategy in vivo. Small-molecule p53 reactivators have been explored in clinical trials [27], although their utility for the treatment of glioma has not been investigated. We hypothesized that in vivo delivery of the mutant p53 small-molecule reactivator, PRIMA-1^Met^ would restore wild-type p53 in mutant p53 (R273C) tumor cells (GBM22) by affecting p53′s canonical role in tumor growth, as well as the non-canonical role regarding xCT expression and function. To overcome limitations of drug delivery, we examined this approach treating mice bearing gliomas in their flank, which allowed for daily tumor growth monitoring. Upon implantation, tumor growth was routinely monitored, and animals were blindly sorted into either a saline or PRIMA-1^Met^ treatment group following a 5-days-on, 2-days-off treatment schema illustrated in Figure 5A. Upon cessation of the experiment, we witnessed a decrease in tumor volume over time for PRIMA-1^Met^-treated animals compared to saline-treated controls (Figure 5B). Furthermore, the growth rate of PRIMA-1^Met^-treated animals was significantly reduced compared to saline-treated control animals (Figure 5C). Consistent with the published function of PRIMA-1^Met^ [28,29], tumors from PRIMA-1^Met^-treated animals exhibited a decrease in p53 protein (Figure 5D). Moreover, PRIMA-1^Met^-treated animals also exhibited a decrease in xCT protein expression (Figure 5E). We also attempted a similar PRIMA-1^Met^ experiment with an intracranial GBM22 tumor model (Appendix A); however, we did not achieve a significant reduction in xCT protein (Appendix A), most likely due to the inability of PRIMA-1^Met^ to penetrate the blood–brain barrier [28].

## 4. Discussion

With a dire need for new GBM therapeutic avenues, SXC has become an attractive target due to its involvement in tumorigenesis [30], tumor-associated epilepsy [6,8], and chemotherapy resistance [30,31]. The FDA-approved SXC antagonist, SAS, has shown promise related to glutamate biology in preclinical glioma studies and small pilot studies with human patients [6,8,10]. Although repurposing SAS for GBM patients would reduce the cost and approval time for a new cancer therapeutic [32], SAS has a short half-life and how much enters the brain remains unknown [9]. Due to these bioavailability hurdles, companies and other academic laboratories have ventured to design new potent lead molecules [33]. Thus, the GBM patient population could benefit from the identification of additional, novel pathways that regulate SXC. Other groups have identified *SLC7A11* transcriptional regulators such as activating transcription factor 3 (ATF3) [34] and p53 [12] in ferroptosis, but little has been explored in the context of glioma. This motivated us to investigate and identify whether p53 was a transcriptional suppressor of *SLC7A11* in GBM.

Since its discovery in the 1970s, p53 has been identified as the guardian of the genome, because its inactivation is pivotal in tumor formation [35]. p53′s official role is a master regulator of the cell cycle and apoptosis cell death pathway; however, researchers have elucidated other non-canonical p53 functions in the cell [36]. For example, Jiang et al. demonstrated that p53 regulates an iron-dependent, non-apoptotic cell death pathway called ferroptosis [12]. Interestingly, during ferroptosis, p53 suppresses *SLC7A11* in order to control cystine metabolism and reactive oxygen species in tumor cells. Furthermore, the ferroptosis inducer, erastin, is a potent xCT inhibitor [12,37]. Other groups have proposed that mutant p53 accumulation negatively regulates *SLC7A11* in non-brain tumor cell lines [38], although our data support the hypothesis that wild-type p53, and not null or mutant p53 PDX GBM tumor cells, possess downregulated *SLC7A11* (Figure 1). Furthermore, our study utilized PDX cells that have naturally occurring p53 mutations and corresponding *SLC7A11*/xCT expression levels, unlike previous studies that had artificially over-expressed mutant p53 in non-brain tumor cell lines [38].

In this study, we demonstrated a molecular relationship whereby SXC expression was inversely related to p53.Wild-type p53 PDX GBM tumor cells have lower *SLC7A11* transcript and xCT protein levels relative to tumors with null or mutant p53 (Figure 1 and Figure 2). Most importantly, through ChIP assays, we determined that a physical association exists between p53 protein and the promoter region of *SLC7A11* (Figure 3A). Through block and mimic experiments, we altered SXC expression and function. *SLC7A11* transcript increased after p53 knockdown (Figure 3B,C) although we did not see a change in xCT protein or glutamate release (Appendix A). Results in the literature suggest that SXC has a long half-life; xCT protein was still abundant in T98G GBM cells 36 h after cyclohexamide treatment [26]. Therefore, although we assessed xCT protein levels 96–120 h post-transfection, it is possible that the transience of siRNA and long xCT half-life made it difficult to manipulate xCT protein expression in this experiment. Conversely, we observed a decrease in xCT protein and glutamate release after p53 over-expression in p53-null PDX cells (Figure 4). Therefore, these studies support p53 as a molecular regulator of SXC in GBM tumor cells.

Although p53 was discovered decades ago as an important tumor suppressor in the pathogenesis of cancer, clinicians cannot genetically change somatic p53 mutations in patients. Therefore, mutant small-molecule p53 reactivators have become attractive candidates in the field of cancer biology [27]. p53 restoration therapies are proving to be viable strategies in tumor management, including those in clinical trials for myeloid neoplasms, and esophageal and ovarian cancers [27]. Among these agents is PRIMA-1^Met^, a pro-drug which is hydrolyzed to the active Michael acceptor, MQ, allowing mutant p53 to undergo a structural change to confer wild-type p53 biology [28] and thermostabilization [39]. Our final in vivo study suggests that PRIMA-1^Met^ is a promising therapeutic for GBM patients, specifically through xCT biology. This in vivo PRIMA-1^Met^ flank experiment showed a reduction in the canonical role of p53, in addition to a decrease in xCT protein (Figure 5). p53 regulates components of the cell cycle, although it also controls the expression of Mdm2 for its own regulation [40]. Therefore, the reduction in p53 protein after PRIMA-1^Met^ treatment supports the published function of this mutant small-molecule p53 reactivator [28,29].

A large focus in glioma has been on xCT-mediated glutamate excitotoxicity, but the antiporter also imports cystine, which is a precursor to the antioxidant, glutathione. Therefore, xCT is a double-edged sword in tumor management and burden because it releases high levels of glutamate, which kills surrounding brain cells, and it can protect itself from therapies through the production of glutathione. Lambert et al. demonstrated that PRIMA-1′s MQ metabolite can react with thiols, and that the inhibition of glutathione synthesis increased tumor suppression mediated through PRIMA-1 [28]. Mandal et al. reported that over-expression of the System Xc^-^ catalytic subunit, xCT, rescues glutathione deficiency [41]. Furthermore, GSH concentrations decrease in neuroblastoma cell lines after 6 h of PRIMA-1 treatment in vitro [42]. The SXC inhibitor also synergizes with PRIMA-1 in vitro to lower GSH levels in non-small lung cancer H2199 cells with mutant p53 [38]. These studies, and ours, suggest that PRIMA-1^Met^ may be an attractive candidate for targeting the intracellular effects of xCT by impairing cancer stem cell antioxidant defense mechanisms [43]. However, although we did use the more lipophilic, methylated version of this small molecule, our intracranial study (Appendix A), in combination with previous work [28], suggests that PRIMA-1^Met^ may have poor blood–brain barrier permeability. Given the possibilities to more effectively advance pharmacotherapy for GBM, such as focused ultrasound and nanoparticle encapsulation [44], PRIMA-1^Met^ formulation and delivery should be considered in future pre-clinical GBM studies.

## 5. Conclusions

Personalized medicine has become an attractive platform for the future of cancer management. Clinicians and scientists have classified GBM tumors into subsets to acknowledge tumor heterogeneity and molecular characteristics of tumor pathophysiology and drug response [45]. Given the results from our previous study with SXC [6] and this current study’s results on xCT regulation via p53, our data may educate choices when it comes to tumor management. For example, the proneural GBM subclass contains a higher percentage of *TP53* mutations; therefore, our study suggests that *SLC7A11* could be an important target gene for proneural mutant *TP53* GBM patients [46]. Our study supports the suggestion that p53 regulation of *SLC7A11* is a novel signaling axis driving GBM tumor biology, as illustrated in the graphical abstract. Therefore, these studies elucidate how targeting the non-canonical relationship between p53 and SXC regulation may provide glioblastoma patients with an innovative clinical avenue and satisfy a largely unmet need for novel therapeutics.

## Figures and Tables

**Figure 1 cancers-13-06169-f001:**
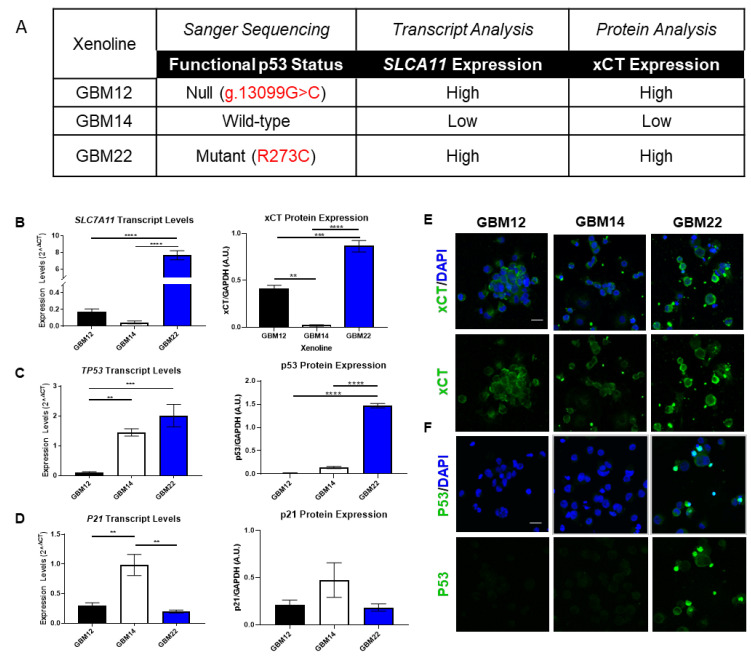
System x_C_^-^ expression is inversely correlated with p53 functional status. (**A**) Sanger sequencing was performed on three different PDX GBM lines. Sequencing revealed different p53 statuses whereby GBM12 cells were p53-null, GBM14 cells were p53 wild-type, and GBM22 cells were mutant p53 (R273C). Transcript analysis and protein analysis results are also summarized in this table from experiments completed in the subsequent panels. (**B**) RT-qPCR and Western blot analysis of *SLC7A11* and xCT expression in all three xenolines from (**A**). (**C**) RT-qPCR and Western blot analysis of *TP53* and p53 expression in all three xenolines from (**A**). (**D**) RT-qPCR and Western blot analysis of *P21* and p21 expression in all three xenolines from (**A**). All RT-qPCR and Western blot experiments are represented by at least three biological replicates. An ordinary one-way ANOVA with Tukey’s multiple comparisons test was performed. Error bars represent the mean ± standard error of the mean (S.E.M.). **** *p* < 0.0001, *** *p* < 0.001, ** *p* < 0.01. (**E**,**F**) All three xenolines from (**A**) were subjected to ICC analysis for xCT (**E**) and p53 (**F**) (green) and counter-stained with nuclear DAPI (blue). Note the characteristic accumulation of mutant p53 in the nucleus of GMB22 cells ((**F**), right panels). Scale bar = 20 µm.

**Figure 2 cancers-13-06169-f002:**
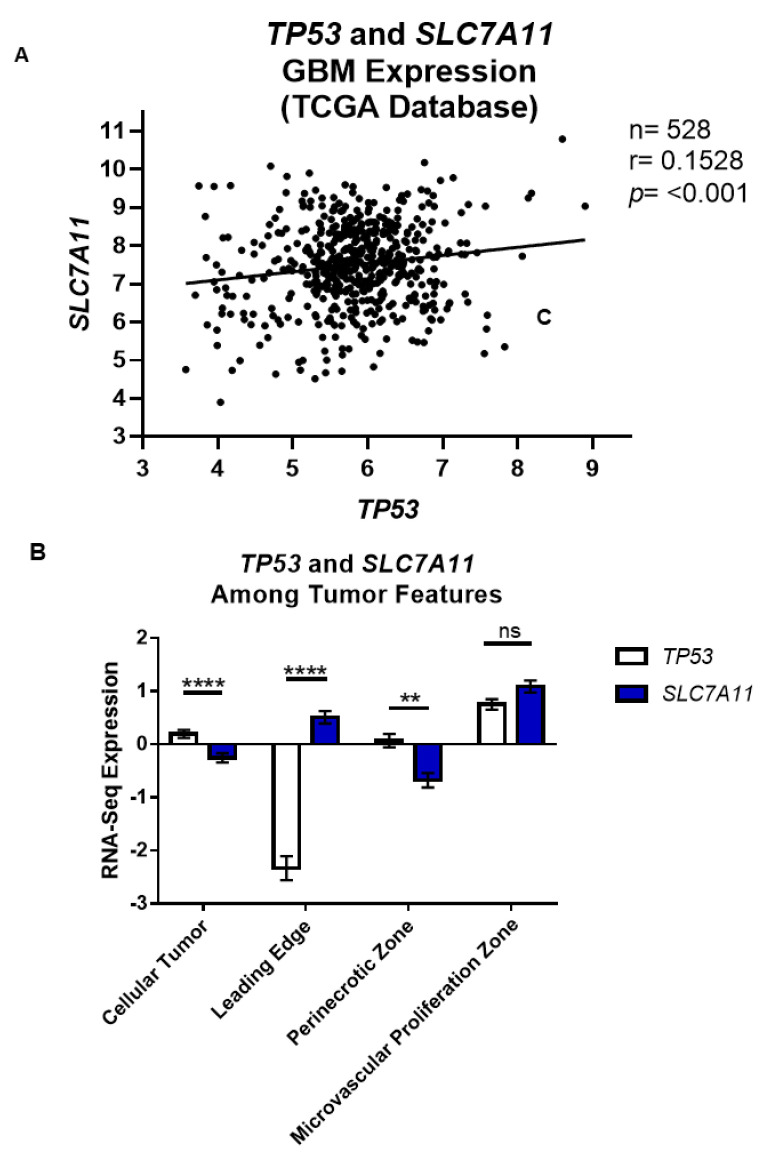
*SLC7A11* expression is inversely related to p53 expression in patient database samples. (**A**) Correlation analysis between *TP53* and *SLC7A11* expression was performed using the GlioVis portal and the TCGA database. (**B**) A gene search for “TP53” and “SLC7A11” demonstrate various features such as the “Leading Edge”, “Infiltrating Tumor”, “Cellular Tumor”, “Perinecrotic Zone”, or “Microvascular proliferation”. These features display various relationships between *TP53* and *SLC7A11*. Two-way ANOVA, with a multiple comparisons post hoc Šidák’s test was performed. Error bars are graphed with the mean ± S.E.M. **** *p* < 0.0001, ** *p* < 0.01.

**Figure 3 cancers-13-06169-f003:**
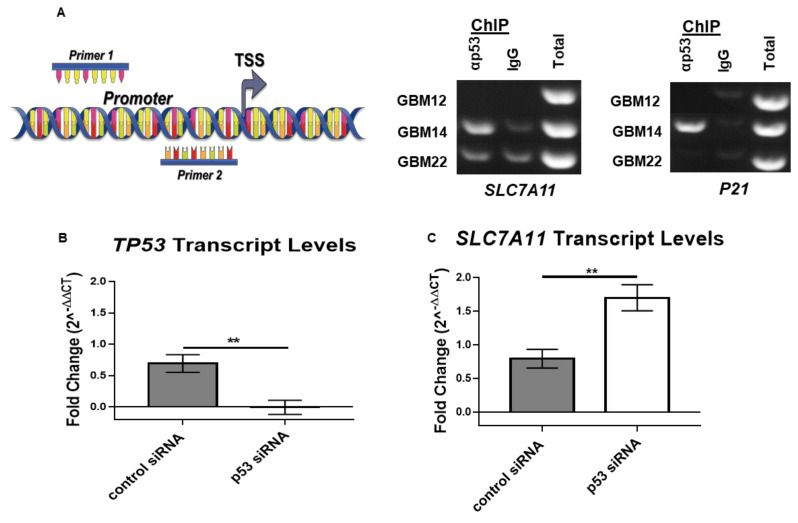
p53 is a molecular switch for xCT expression and function. (**A**) A graphic summary of primer location for ChIP experiments. Gels show amplification of *SLC7A11* and *P21* after ChIP in three PDX lines. In wild-type p53 tumor cells (GBM14), there is a clear band for *SLC7A11* and *P21* over the IgG control, unlike in p53-null (GBM12) and mutant (GBM22) lines. Multiple ChIP experiments were performed; this is a representative gel. (**B**,**C**) All RT-qPCR experiments represent at least three biological replicates. An unpaired two-tailed t-test was performed, and error bars represent the mean ± S.E.M. ** *p* < 0.005.

**Figure 4 cancers-13-06169-f004:**
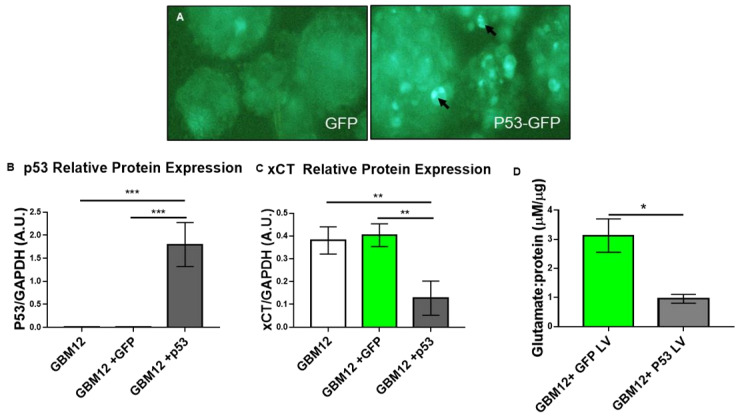
p53 over-expression decreases in vitro GBM xCT expression and glutamate release. (**A**) Fluorescent images were acquired with a Nikon Ti2 microscope to document the successful infection of GBM12 cells with either a lentivirus over-expressing GFP (left) or p53 conjugated to GFP (right); 10x magnification. Densely expressed GFP in the P53-GFP over-expressing tumor cells (arrows). (**B**,**C**) Western blot analysis of transduced tumor cells shows P53 overexpression in GBM12 cells (**B**) reduces high xCT expression (**C**) compared to non-transduced and GFP controls. All experiments represent at least three biological replicates. An ordinary one-way ANOVA with Tukey’s multiple comparisons test was performed. Error bars represent the mean ± S.E.M. *** *p* < 0.005, ** *p* < 0.01 (**D**) Media were taken from transduced cells analyzed in (**B**,**C**) and the glutamate content was measured. p53 over-expression reduced glutamate release in GBM12 cells. An unpaired, two-tailed *t*-test was performed. Error bars represent the mean ± S.E.M. Experiments represent at least three biological replicates. * *p* < 0.05.

**Figure 5 cancers-13-06169-f005:**
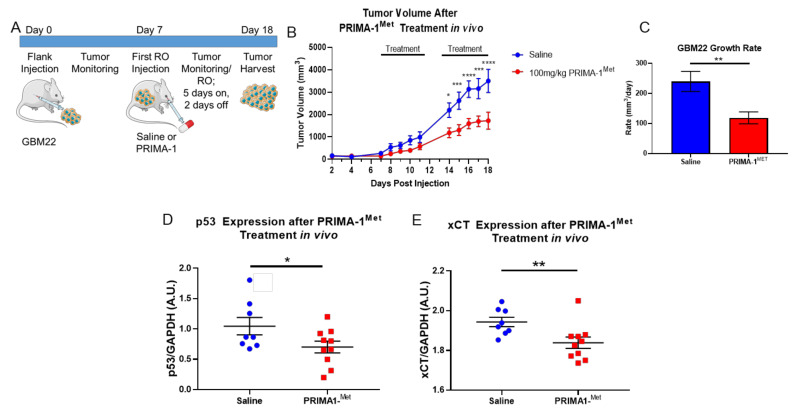
Mutant p53 chemical restoration mitigates SXC-mediated glioma biology in vivo. (**A**) A schematic representing the tumor implantation and dosing regimen for our PRIMA-1^Met^ in vivo flank experiment. There were 8–10 animals per dosing group. (**B**) Tumor volumes were measured in animals during routine animal health monitoring. Tumor growth decreased in the PRIMA-1^Met^-treated animals compared to saline controls. A two-way ANOVA with Šidák’s multiple comparisons test was performed. Error bars represent S.E.M. **** *p* < 0.0001, *** *p* < 0.001, * *p* < 0.05. (**C**) The rate of tumor growth from (**B**) plotted for both treatment groups also shows that the growth rate decreased in the PRIMA-1^Met^-treated animals compared to saline controls. An unpaired t-test was performed. Error bars represent the mean ± S.E.M. ** *p* < 0.005 (**D**,**E**) Western blot analysis of harvested tumor tissue was performed after the conclusion of the dosing experiment. Both p53 and xCT protein levels were decreased in PRIMA-1^Met^-treated animals compared to saline controls. An unpaired, one-tailed t-test was performed. ** *p* < 0.01, * *p* < 0.05.

## Data Availability

The data presented in this study are available in this article (and Appendix A).

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
