# Peer review of "Transcriptional Regulation of Amino Acid Transport in Glioblastoma Multiforme"

_cancers, 2021, doi:10.3390/cancers13246169_

Round 1

Reviewer 1 Report

     The current manuscript demonstrates that wild-type p53 is a transcriptional repressor of SLC7A11, the gene encoding the System xc- (SXC) catalytic subunit, xCT, in GBM. Using patient derived (PDX) tumor xenolines with wild-type, mutant,  and null p53, they provide evidence that xCT expression is inversely correlated with p53 expression in patient tissue. Their chromatin immunoprecipitation studies further reveal that p53 binds to the SLC7A11 promoter, and suppresses gene expression in PDX GBM cells. Importantly, providing a mechanistic approach and not merely a correlative one, they include results of genetic knockdown of p53, which they show increases SLC7A11 transcript levels, and alternatively, show that p53 over-expression in p53 null GBM cells can down-regulate xCT expression and glutamate release. Further, the authors effectively demonstrate that treatment of mice xenografted with flank gliomas with the mutant p53 reactivator, PRIMA-1Met, reduces tumor growth associated with reduced xCT expression. The paper is well written, the content is comprehensive and persuasive, and the major points are stated clearly and well supported. This study is important to the GBM field and demonstrates that manipulating p53 in tumors alters SXC expression and function, and illustrates the potential therapeutic utility of mutant p53 reactivator for treatment of GBM.

The following minor revisions/ suggestions would increase the support for the authors’ conclusions:

1) To ensure that the results the authors’ see are not cell-line specific, their data would need to be reproduced with at least another PDX cell line for each of the groups (wild-type, mutant,  and null p53), given that the authors have access to a characterized comprehensive set of patient derived glioma PDX lines. In the current paper, they only show results with one cell line per group (GBM12 (null), GBM14 (wild-type), and GBM22(mutant);

2) Figure 6 is mentioned in the Conclusions (line 503), but is missing and needs to be corrected, or the figure added; including this schematic (Figure 6) showing p53 regulation of SLC7A11 as a novel signaling axis driving GBM tumor biology could definitely improve the manuscript.

Author Response

Reviewer #1:

1) To ensure that the results the authors’ see are not cell-line specific, their data would need to be reproduced with at least another PDX cell line for each of the groups (wild-type, mutant, and null p53), given that the authors have access to a characterized comprehensive set of patient derived glioma PDX lines. In the current paper, they only show results with one cell line per group (GBM12 (null), GBM14 (wild-type), and GBM22(mutant);

Response: These PDX lines were from naturally occurring tumors and therefore we are not privileged to another set of PDX lines that possess the exact same point mutations for comparison (e.g. R273C in GBM 22 and the point mutation causing resulting in p53 null in the GBM12 line).

2) Figure 6 is mentioned in the Conclusions (line 503), but is missing and needs to be corrected, or the figure added; including this schematic (Figure 6) showing p53 regulation of SLC7A11 as a novel signaling axis driving GBM tumor biology could definitely improve the manuscript.

Response: Thank you for pointing out this formatting error. Figure 6 was meant to be a graphical abstract and was in a different version of the manuscript that was accidentally not uploaded. This abstract has now been copied and pasted into the body of the text with its corresponding figure legend.

Reviewer 2 Report

The authors demonstrate a link between p53 and xCT and demonstrate that p53 reactivation is a potential line of therpay for glioblastoma.

My main critique of this work is that all the gene expression changes are measured semi-quantitatively. I do not understand why the authors did not use a quantitative method that is much more accurate and convincing.

Author Response

General: Another round of spellcheck and review was performed for grammar.

My main critique of this work is that all the gene expression changes are measured semi-quantitatively. I do not understand why the authors did not use a quantitative method that is much more accurate and convincing.

Response: The ultimate goal of this work was to demonstrate whether p53 as a transcription factor would regulate xCT at a functional protein level. Therefore, most of our desired readouts were via Western blot or glutamate release to measure these outcomes.

Reviewer 3 Report

Here Umans et al demonstrated the negative correlation of p53 with xCT and its gene SLC7A11 in Glioblastoma (GBM) cancer using patient-derived cells expressing WT, null or mutant (overexpress) p53. After performing Sanger sequencing to clarify p53 state in patient GBM cells, the authors evaluated transcript and protein levels at baseline, after siRNA knockdown study or after lentiviral-mediated overexpression. Besides, the authors tested p53 reactivator small molecule PRIMA1Met in mice model bearing GBM xenograft on its flank or intracranial, showing therapeutic efficacy in the former model. Overall, I think the authors have presented quite a significant work and a nice article for publication in Cancers, subject to few clarifications or revisions below: - Several recent articles (DOI: 10.1158/1538-7445.AM2016-4357; Nat Commun 8, 14844 (2017). https://doi.org/10.1038/ncomms14844; Cells 2021, 10(1), 108; https://doi.org/10.3390/cells10010108) have discussed the inverse relationship of p53 and xCT in other cancers. Given its close relevance, discussion and/or comparison with these articles is of importance. - In the manuscript text, the authors mention "significant albeit small inverse relationship between TP53 & SLC7A11". However, figure 2A shows an increasing linear correlation. Please clarify this - Suggest the authors to explain the reason for drop in p53 protein after PRIMA-1Met treatment in vivo. - Did the authors look at PRIMA-1 accumulation at tumor site (at least for flank tumor setting)? While significant, the reduction in tumor growth rate is only ~50% despite large daily dose of 100mg/kg injection. - Please correct this reference issue: ref 12 & 17 are duplicates.

Author Response

Several recent articles (DOI: 10.1158/1538-7445.AM2016-4357; Nat Commun 8, 14844 (2017). https://doi.org/10.1038/ncomms14844; Cells 2021, 10(1), 108; https://doi.org/10.3390/cells10010108) have discussed the inverse relationship of p53 and xCT in other cancers. Given its close relevance, discussion and/or comparison with these articles is of importance.

Response: The citation Nat Commun 8, 14844 (2017) was already referenced and discussed in the manuscript (citation #36). We have included the Cells 2021, 10(1), 108 reference review article in the discussion of the manuscript.

In the manuscript text, the authors mention "significant albeit small inverse relationship between TP53 & SLC7A11". However, figure 2A shows an increasing linear correlation. Please clarify this

Response: The line which is fitted to the data in Figure 2A is linear, although the data itself is not perfectly linear. This is indicated by r of 0.1528, while positive is not truly linear which would be a value of 1.0. As mentioned in the following sentence, “Note that this data may be skewed by misrepresentation of true mutational status of p53 when relying on pathological IHC rather than Sanger sequencing”. While we see a correlation in our pure PDX lines that had a mutational status determined by sequencing, the samples in this patient database had p53 status determined by IHC, a less reliable method that we have previously investigated (Roshandel, A.K.  et al, Oncotarget, 2019 DOI: 10.18632/oncotarget.27252)

Suggest the authors to explain the reason for drop in p53 protein after PRIMA-1Met treatment in vivo.

Response: Citations 26 and 27 were accidentally superscripted on line 407. These references (doi:10.1111/joim.12336 and doi:https://doi.org/10.1016/j.ccr.2009.03.003) infer how that the proper folding of wild-type p53 from its mutant form can now self-regulate itself by regulating other molecular players such as MDM2. This is also mentioned in the discussion “While p53 regulates components of the cell cycle, it also regulates the expression of Mdm2 for its own regulation [38]. Therefore, the reduction in p53 protein after PRIMA-1Met treatment, supports the published function of this small molecule mutant p53 reactivator [26,27].

Did the authors look at PRIMA-1 accumulation at tumor site (at least for flank tumor setting)? While significant, the reduction in tumor growth rate is only ~50% despite large daily dose of 100mg/kg injection.

Response: We did not have the capabilities in this designed study to assess PRIMA-1 accumulation at the flank tumor site. We needed to collect the tumor for tissue processing. In addition, our treatment dose in mg/kg was similar to another efficacious flank tumor study using PRIMA-1 (Bykov, V et al, Nat 2002, DOI: 10.1038/nm0302-282) although with less frequency to ensure humane treatment of the animals. They were receiving retro-orbital injections in alternating eyes to allow for recovery. The treatment paradigm was 5 days on, 2 days off, and 5 days on before tissue collection. Because of the fast growth rate of the tumor and the 2 days off in treatment, we hypothesize the growth rate wasn’t diminished further due to these reasons.

Please correct this reference issue: ref 12 & 17 are duplicates.

Response: Thank you for pointing out the duplication in these references. This has been resolved in the References library.